# Homologous Recombination Deficiency Scar: Mutations and Beyond—Implications for Precision Oncology

**DOI:** 10.3390/cancers14174157

**Published:** 2022-08-27

**Authors:** Alexander M. A. van der Wiel, Lesley Schuitmaker, Ying Cong, Jan Theys, Arne Van Hoeck, Conchita Vens, Philippe Lambin, Ala Yaromina, Ludwig J. Dubois

**Affiliations:** 1The M-Lab, Department of Precision Medicine, GROW—School for Oncology and Reproduction, Maastricht University, 6229 ER Maastricht, The Netherlands; 2Center for Molecular Medicine and Oncode Institute, University Medical Center Utrecht, 3584 CG Utrecht, The Netherlands; 3Institute of Cancer Science, University of Glasgow, Glasgow G61 1BD, Scotland, UK; 4Department of Radiation Oncology, The Netherlands Cancer Institute, 1066 CX Amsterdam, The Netherlands

**Keywords:** cancer, DNA repair, homologous recombination, homologous recombination deficiency, homologous recombination deficiency scar, biomarkers, precision oncology

## Abstract

**Simple Summary:**

A characteristic across several cancer types is a homologous recombination deficiency (HRD). HRD is associated with a better response to several anticancer therapies. Adequate assessment of HRD can therefore improve the outcome of such therapies. However, current methods to assess HRD are insufficient, leading to an underestimation of patients with HRD. This review discusses more accurate methods to detect HRD and how these can be applied for more personalized cancer treatment.

**Abstract:**

Homologous recombination deficiency (HRD) is a prevalent in approximately 17% of tumors and is associated with enhanced sensitivity to anticancer therapies inducing double-strand DNA breaks. Accurate detection of HRD would therefore allow improved patient selection and outcome of conventional and targeted anticancer therapies. However, current clinical assessment of HRD mainly relies on determining germline *BRCA1/2* mutational status and is insufficient for adequate patient stratification as mechanisms of HRD occurrence extend beyond functional BRCA1/2 loss. HRD, regardless of *BRCA1/2* status, is associated with specific forms of genomic and mutational signatures termed HRD scar. Detection of this HRD scar might therefore be a more reliable biomarker for HRD. This review discusses and compares different methods of assessing HRD and HRD scar, their advances into the clinic, and their potential implications for precision oncology.

## 1. Introduction

The homologous recombination (HR) pathway involves a myriad of mediators—including BRCA1, BRCA2, ataxia telangiectasia mutated (ATM) kinase, RAD51 paralogs, and RAD52—and is essential for high-fidelity repair of double-strand DNA breaks (DSBs) [1]. HR deficiency (HRD) is prevalent (17.4%) in several cancer types, including breast, ovarian, pancreatic, and prostate cancer [2], and is associated with increased sensitivity to DNA-damaging agents, such as platinum-based chemotherapeutics, and inhibitors of poly (ADP-ribose) polymerase (PARP) [3,4]. Accurate detection of HRD is therefore clinically relevant as it could improve patient selection and outcome of conventional and targeted therapies.

In the current clinical setting, assessing germline or somatic *BRCA1/2* mutational status is mainly used to determine HRD [5]. However, mechanisms of HRD occurrence extend beyond the functional loss of BRCA1/2 [6,7], including mutations in other HR-related genes, epigenetic silencing and changes in gene expression of these genes, or by other unknown mechanisms, rendering germline testing insufficient for adequate patient selection and underscoring the need for better biomarkers of HRD. As HRD tumors exhibit a higher degree of genomic instability and are characterized by specific mutational footprints [8,9,10,11,12], detection of these phenotypic consequences of HRD—termed HRD scar—can thus provide a better tool to detect HRD regardless of its cause.

The current review provides an overview of different methods and biomarkers to detect HRD and HRD scar, compares their advantages and limitations, and discusses their advances into the clinic with potential implications for precision oncology.

## 2. The Homologous Recombination Repair Pathway

The two major pathways for repair of DSBs—essential for maintenance of genomic integrity—are nonhomologous end joining (NHEJ) or HR [13]. NHEJ is a highly flexible and fast-acting DNA repair pathway active mainly in the G1 phase of the cell cycle but also contributes to DSB repair throughout the entire cell cycle, ligating two broken DNA strands with minimal regard to sequence homology [14]. Junctions repaired by NHEJ, however, often harbor mutations or loss of genomic information. HR, on the other hand, is a process of slower kinetics using a homologous DNA sequence as a template for highly accurate repair of DSBs [13]. Unlike NHEJ, HR-mediated repair is mainly restricted to the S phase and G2 phase of the cell cycle [15], when sister chromatids are more easily accessible. HR is a multistep process extensively reviewed by others [16,17]: after recognition of a DSB by the MRE11-RAD50-NBS1 (MRN) complex, ATM is activated. The replication protein (RPA) is recruited and coats the lesion, which in turn activates ataxia telangiectasia and RAD3-related (ATR) kinase. RPA is subsequently replaced by RAD51 mediated by BRCA2, which itself is recruited to the DSB by BRCA1 and partner and localizer of BRCA2 (PALB2). The resulting RAD51-DNA nucleoprotein subsequently initiates homology search followed by displacement loop (D-loop) formation and strand invasion, a process in which RAD52 is involved [18]. DNA synthesis starts and can be mediated via distinct HR subpathways.

As becomes evident, HR is a complex pathway engaging a multitude of sensors and mediators. Despite BRCA1/2 being centrally involved in HR, a deficiency in any other element of HR can also disrupt HR, effectively leading to HRD. Supportive of this, deficiencies in non-BRCA HR-related genes have indeed shown to result in heightened sensitivity to platinum-based agents and PARP inhibitors (PARPis). For example, patients with a *PALB2* mutation significantly benefit from cisplatin [19] and olaparib treatment [20]. Interestingly, no benefit of olaparib treatment was observed in *ATM*-mutated breast [20] and prostate cancer patients [21]. Nonetheless, and as stated above, these findings underscore the inadequacy of assessing germline or somatic *BRCA1/2* status in identifying patients with HRD. Assays focusing on the phenotypical or functional consequences of HRD, regardless of its cause, may prove to be better biomarkers of HRD.

## 3. Large-Scale Genomic Aberrations Associated with HRD

Large-scale genomic aberrations were initially detected in patients harboring autosomal hereditary recessive mutations in genes encoding proteins orchestrating and regulating HR. Ataxia telangiectasia was one of the first DNA damage response disorders identified [22], a syndrome associated with genomic instability, cancer predisposition, and increased sensitivity to therapies inducing DBSs [23]. A different example is the RecQ family of DNA helicases that play an essential role in HR-mediated repair of DSBs [24] and maintenance of genomic integrity [25]. Mutations in members of the RecQ family have been associated with Bloom’s syndrome [26], Werner’s syndrome [27], and the Rothmund–Thomson syndrome [28]. These syndromes are characterized by large-scale genomic aberrations that were later found to be caused, indirectly, by defective HR [28].

Indeed, in the absence of HR due to HRD, nonconservative, i.e., less precise, forms of DNA repair such as NHEJ are used to repair DNA damage [7]. This error-prone DNA repair ultimately leads to specific large-scale genomic aberrations or scars, including telomeric allelic imbalance (TAI), loss of heterozygosity (LOH), and large-scale transitions (LST) [29] (Figure 1). Detection and quantification of these genomic scars—most often using single-nucleotide polymorphism (SNP)-based assays—can thus identify tumors as HRD regardless of its underlying cause.

A first large-scale genomic aberration found to be associated with HRD was TAI, defined as regions of allelic imbalance extending to the telomeric regions but not crossing the centromere. High degrees of TAI have shown to be inversely associated (R = −0.50; *p* = 0.0053) with *BRCA1* expression and to successfully predict (AUC = 0.74) the response to cisplatin of triple-negative breast cancer (TNBC) patients in two phase 2 clinical trials (NCT00148694 and NCT00580333) [30]. Importantly, this association remained significant when only *BRCA1/2* wild-type cases were included, underscoring that causes of HRD extend beyond functional loss of BRCA1/2. The majority of TAI regions occur within 25 kB of copy-number variations (CNVs), suggesting that CNV-associated mechanisms such as increased replicative stress or stalled replication forks might be underlying mechanisms of TAI [30].

LOH was identified later as another genomic aberration associated with HRD. LOH is defined as the loss of one of two alleles at a specific locus, either by deletion of this allele (copy-loss LOH) or by deletion of the allele accompanied by duplication of the other allele (copy-neutral LOH) [31]. Depending on its underlying cause, regions of LOH of different length and pattern can eventuate. Abkevich and colleagues discovered that LOH of intermediate lengths (>15 Mb, but less than the entire chromosome) were significantly associated with HRD, defined as a deficiency in *BRCA1*/2 (*p* = 10^−11^) or *RAD51C* (*p* = 0.0003), in ovarian cancer [8]. Similar to BRCA1/2, RAD51 plays a central role in HR and is involved in the homology search and strand invasion steps [32]. Importantly, a substantial fraction of *BRCA1/2* and *RAD51C* wild-type tumors also exhibited high degrees of LOH [8], indicative of mechanisms of HRD beyond mutations in these HR genes. Indeed, no differences between the extents of LOH were observed between HRD caused by mutations and either promoter methylation or low gene expression levels [8]. Later clinical trials used a prespecified cutoff value of 14% or more genomic LOH to classify tumors as LOH^high^, or HRD [33].

As a third characteristic of genomic instability associated with HRD, Popova and colleagues found the number of LSTs, defined as a chromosomal break between flanking regions larger than 10 Mb, to be a robust indicator of *BRCA1/2* mutational status in breast cancer patients [9]. A tumor displaying ≥15 or ≥20 LSTs, if near-diploid or near-tetraploid, respectively, was classified as HRD. The majority of these LSTs corresponded to interchromosomal translocations and were associated with increased (*p* < 10^−16^) GC-content and gene-rich chromosomal regions [9]. A subsequent study employing 456 breast cancer patients of varying subtypes validated that, in addition to *BRCA1/2* status, high degrees of LST were correlated to *RAD51C* mutational status [34]. In a The Cancer Genome Atlas (TCGA) cohort of 467 breast cancer patients, a significant association between the degree of LST and *BRCA1/2* and *RAD51C* mutational status or expression levels was also found [34]. Importantly, 29% of LST^high^ cases did not show any alteration in *BRCA1/2* or *RAD51C* genes [34], again suggestive of other underlying causes of HRD.

Although all three genomic scars are highly correlated with each other in certain cancer types such as breast cancer [35,36] and can individually predict *BRCA1/2* mutational status, the combination of TAI, LOH, and LST parameters was found to perform best at distinguishing HR proficient from deficient tumors [29,37]. Indeed, Telli and colleagues demonstrated that the HRD score, i.e., the unweighted numeric sum of TAI, LOH, and LST, exhibited superior performance at detecting HRD in breast cancer patients [4]. Using a training dataset comprising 1058 breast and ovarian tumors of which 268 were *BRCA1/2* deficient, an HRD score of ≥42 was set as threshold to identify HRD tumors with a sensitivity of 95% [4]. A positive HRD score was validated to be associated with *BRCA1/2* mutations and with response to platinum-based agents in the PrECOG 0105 breast cancer cohort (NCT00813956) even in the absence of *BRCA1/2* mutations [4]. The HRD score has since been further validated in other cancer types [38,39,40], and has successfully been used in clinical trials for patient selection as recently reviewed extensively by others [41]. Importantly, however, it should be noted that the threshold of HRD score—and perhaps the HRD score itself—is probably dependent on the tumor type and should be evaluated or optimized carefully before clinical application.

Lastly, it should be mentioned that other large-scale, genome-wide aberrations have also been associated with HRD. Chromothripsis, for example, is a unique form of genomic instability defined as a single, cataclysmic event whereby a single or a few chromosomes are affected by tens to thousands of clustered rearrangements [42]. Higher degrees of chromothripsis were found in HRD tumors, as evidenced by a strong link with *ATM* mutations in patients suffering from acute lymphoblastic leukemia [43]. However, the precise underlying mechanisms causing chromothripsis remain unclear [44] and should be further extensively investigated to unveil the link with HRD, and allow application for detection of HRD in a clinical setting.

## 4. Mutational Signatures of HRD

In cancer, over 100 different mutational signatures—comprising single and multiple base substitutions, small insertions and deletions, small genomic rearrangements and chromosome CNVs—have been described, caused by various endogenous and exogenous factors such as mutagen exposures, replication errors, and deficient DNA maintenance [45,46]. These mutational signatures are patterns of somatic passenger mutations that arise during tumorigenesis and can thus provide insights into the underlying causes of individual cancers [47]. Every signature serves as an imprint of the past and ongoing distinct DNA damage and repair mutation processes in the tumor [48]. Mutational signatures are divided into four categories: single-base substitution (SBS), double-base substitution (DBS), small insertion/deletion (ID), and rearrangement signatures (RS). Historically, SBSs are classified based on the flanking sequence context of each possible substitution type, resulting in a 96-channel pattern [12], DBSs are defined by 78 channels [49], and RS are classified using a 32-channel system [12]. IDs, on the other hand, are more recently defined and remain relatively underexplored [47]. IDs are classified according to a set of 83 channels, and to date, 17 ID signatures have been identified [49]. In context of mutational signatures associated with HRD, SBSs have been most extensively investigated.

Single-base substitution Signature 3 (SBS3)—a uniform pattern of mutations across all 96 possible substitution types—is one of the mutational signatures discovered to be correlated with HRD—specifically with *BRCA1/2* mutations [12,50]—in breast cancer, and was later extended to pancreatic [51,52], ovarian [53], and gastric cancer [54]. However, a later study demonstrated that most samples in the top quartile of SBS3 activity did not exhibit deleterious *BRCA1/2* mutations, suggesting that other events might also contribute to SBS3 [55]. Indeed, several other causes leading to HRD have since also been associated with elevated SBS3, such as epigenetic silencing of the *BRCA1* promoter region [55], deleterious mutations in *PALB2* [55,56], a crucial component for recruitment of BRCA2 to DSBs, and *RAD51C* promoter methylation [55]. It should be noted that similar but different phenotypes of SBS3 may arise dependent on the underlying cause and even tumor type, e.g., *BRCA2* and *PALB2* inactivation lead to larger deletions when compared to *BRCA1*- and *RAD51*-related HRD [57].

Comparison of SBS and LOH or LST for detection of HRD demonstrated that using SBS3 is more sensitive and could identify up to 11% more tumors as HRD with a yet unknown underlying cause [55]. Rather than SNP-based arrays used to assess LOH, LST, and TAI, detection of SBS3 relies mostly on whole genome sequencing (WGS) or whole exome sequencing (WES) data, potentially explaining this difference in sensitivity. In addition, SBS3 is more sensitive in identifying *BRCA2* and *RAD51C* mutational events [55]. Nonetheless, selecting a cutoff value to discriminate between HRD-proficient and -deficient tumors is not straightforward when using this signature alone, given its limited discriminatory mutation profile potentially resulting in non-HRD SBS mutations wrongly being assigned to SBS3, and because the mutational spectrum of a single signature is often affected by other signatures active in the same dataset. Indeed, besides SBS3, evidence has demonstrated that HRD can give rise to at least five distinct mutational signatures depending on its cause [12,58]. SBS8—characterized by C > A substitutions—has also been associated with HRD, although to a lesser extent than SBS3 [59,60]. ID6 is a microhomology-mediated deletion signature often seen in *BRCA1/2* mutated tumors [49,61], and importantly, has shown to be the most sensitive WGS-related HRD biomarker in the HRD prediction tools HRDetect [62] and CHORD [6]. Lastly, RS3 is elevated in tumors with a *BRCA1*, but not *BRCA2* mutation and is designated by <10 kb tandem duplications [12], whilst RS5—<10 kb deletions—is associated with mutations in both *BRCA1* and *BRCA2* [12]. Presence of these different signatures of the same etiology has shown to impede accurate detection of an SBS3 in a sample [63]. SBS3 is therefore probably not as specific for HRD as previously believed, and use of SBS3 alone to classify tumors as HRD has shown to overestimate HRD in esophagus, lung, head and neck, and uterine cancers [64]. Additionally, current methods of signature detection and analysis are inadequate if the mutational count is low, either due to too few genomes or too few mutations per genome [63].

To address these limitations, a novel computational tool called Signature Multivariate Analysis (SigMA) was developed, using a likelihood-based approach that can detect mutational signatures such as SBS3 even when the mutational count is low [63]. SigMA is an algorithm based on hierarchical clustering of tumor types based on the average entire mutational spectrum, which includes other mutational signatures in addition to SBS3 and varies across and within tumor types [63]. Using a simulated dataset of breast cancer patients, application of SigMA yielded a sensitivity of 74% to identify HRD cases, which was markedly higher than other methods [63]. In the same dataset, use of SigMA doubled the number of cases identified as HRD without inherited mutations [63].

## 5. Integrative Models to Predict HRD

HRD can result in a myriad of genomic scars, ranging from distinct mutational signatures to large-scale genomic aberrations that not only vary with the underlying cause of HRD, but also differ between tumor types. To encompass the full complexity of the genomic scar associated with HRD, models integrating multiple characteristics discussed above have been developed. Currently, the two most commonly used and clinically validated assays to detect HRD and genomic HRD scars are the FDA-approved myChoice^®^ CDx (Myriad Genetics, Inc., Salt Lake City, UT, USAC) and FoundationOne^®^ CDx (F1CDx; Foundation Medicine, Inc., Beverly, MA, USA) assays [41]. To assess HRD, myChoice^®^ CDx utilizes both the combined HRD score [4] of TAI, LOH, and LST, and mutational *BRCA1/2* status. Tumors having HRD scores ≥42 and/or alterations in *BRCA1/2* are considered as HRD-positive. FoundationOne^®^ CDx, on the other hand, includes the percentage of genomic LOH (%LOH > 16) and next-generation sequencing of 315 genes, including HR-related genes extending beyond *BRCA1/2*, and other cancer-related genes such as *NOTCH* and *MTOR* [33,65]. Consequently, these two assays are not equivalent and should thus not be considered interchangeable [66].

More recently developed and yet to be approved for clinical use, HRDetect is a lasso-weighted logistic regression model based on WGS data that includes microhomology-mediated deletions, SBS3, RS3, RS5, HRD score (TAI, LOH, and LST), and SBS8 [62]. In a cohort of 560 patients suffering from breast cancer, HRDetect could identify and, importantly, discriminate between *BRCA1/2*-deficient tumors with a sensitivity of 98.7%, using a probabilistic cutoff of 0.7 [62]. In a validation cohort of 80 breast cancer patients, the sensitivity of HRDetect remained high at 86% [62], emphasizing the value of combining multiple mutational and genomic signatures to predict HRD. Indeed, in comparison to other methods of assessing HRD, especially using individual mutational signatures and genomic scar-based indices as the HRD score, HRDetect performed superior [62]. HRDetect was later validated by Golan et al. in a cohort of 391 patients with pancreatic ductal adenocarcinomas (PDAC) and could predict *BRCA1/PALB2* deficiency with a sensitivity and specificity of 98% and 100%, respectively [67]. Importantly, HRDetect was able to classify an additional 7% to 10% of PDAC patients without known germline mutations as HRD [67]. In addition, in the phase 2 RIO trial (EudraCT 2014-003319-12), HRDetect was more specific in identifying HRD—as defined as mutations or epigenetic alterations in *BRCA1/2*, *PALB2*, or *RAD51C*—in TNBC patients when compared to the HRD score, with up to 69% of patients classified as HRD [68]. The utility of HRDetect in other cancer types, however, remains to be validated: in an ovarian cancer dataset (*n* = 425), HRDetect (AUC = 0.823) did not perform better in identifying HRD than the HRD score (AUC = 0.837) [69].

Using a different approach, the random forest-based model classifier of homologous recombination deficiency (CHORD), employing a pan-cancer approach, was developed [6]. Previous models, such as HRDetect, are developed based on one cancer type necessitating validation on other cancer types before application [69]. CHORD was developed using WGS data of 3584 patients from a pan-cancer metastatic cohort comprised of 20 different cancer types including amongst others pancreas, head and neck, prostate, breast, ovary, colon, and lung cancer [6,70]. CHORD uses a combination of 29 mutational features of three somatic mutation categories: SBSs, IDs, and structural variants (SV), with the presence of deletions with flanking homology of ≥2 bp being the most important predictor of HRD [6]. The features used by CHORD can, in addition to predicting HRD, also distinguish between BRCA1-type HRD and BRCA2-type HRD using 1–10 kb and, to a lesser extent, 10–100 kb duplications. Using a probabilistic cutoff of 0.5 (i.e., the sum of *BRCA1* and *BRCA2* deficiency probability), CHORD can detect HRD with overall low false-positive (<2%) and false-negative (<6%) rates [6]. CHORD was validated and compared with the predictions of HRDetect on two independent datasets, the BRCA-EU breast cancer dataset (*n* = 543) and the pan-cancer analysis of whole genomes (PCAWG) dataset (*n* = 1854), which demonstrated similar performance for both models (AUC = 0.98) [6]. In contrast to HRDetect, however, CHORD does not rely upon mutational signatures, suggesting that accurate prediction of HRD is feasible without additional mutational signature extraction steps [71]. Additionally, as CHORD has been developed and validated on pan-cancer cohorts, it is much better suited for detection of HRD in tumor-agnostic clinical trials.

## 6. Limitations of Detecting Mutational Scars

Depending on the method used to detect mutational or genomic scars associated with HRD, fresh tissue might be required to accurately assess the presence of the respective HRD scar. Routinely used in clinical pathology, formalin-fixed paraffin-embedded (FFPE) tissue samples harbor artifacts that can severely compromise detection of, e.g., SBS3 [62] and is a first limitation of applying HRD scar assessment in clinical setting.

Second, while mutational scar-based methods of detecting HRD identify HRD cases beyond *BRCA1/2* mutations, a principal limitation of detecting mutational and genomic scars is that these represent past events and do not necessarily reflect current HRD status, possibly resulting in false-positive or false-negative classification of a tumor according to HRD status. Reversions or secondary mutations in *BRCA1/2* [72], *RAD51* [73], and *PALB2* [74] have shown to restore HR function and induce resistance to platinum-based agents or PARPis and could therefore lead to a false-positive prediction. On the other hand, recent acquisition of an HRD phenotype could result in a false-negative classification due to insufficient mutational or genomic events for accurate HRD scar detection. Genetic testing of HRD genes should thus be complementary to HRD scar detection providing supporting information for making a final decision on a patients’ HR status. In light with this, assays relying on WGS, although not routinely used (yet) for clinical diagnostics, have the advantage that both genetic testing and HRD scar detection can be performed simultaneously.

Additionally, the mutational scars described above have been mostly defined by comparison with, or have been validated against *BRCA1/2*—or other HR-related genes—mutated samples. Often, however, little evidence is provided that these specific mutations present are actually resulting in HRD, i.e., leading to functional HRD loss. Elaborative of this, assays reflecting real-time, functional HR status might also serve as potential biomarkers. Examples of such assays include evaluation of transcriptional profiles, protein expression levels, or functional assays such as RAD51 foci induction, or sensitivity to platinum-based therapies [5,75,76,77,78,79]. Essers and colleagues developed an RNA expression-based HRD signature, based on cellular crosslinker sensitivity, fully independent of mutational status of HRD genes. This gene expression signature could retrospectively successfully predict response of 180 head and neck cancer patients to cisplatin. However, despite being biologically useful, application of these real-time indicators of HRD is currently not straightforward in the clinic. Nonetheless, integrated functional repair and genomic (scar) analyses could further improve the prognostic and predictive value of genetic biomarkers [80].

## 7. HRD Scar in the Clinic: Implications for Precision Oncology

### 7.1. PARPis

Accurate detection of HRD and subsequent patient selection could drastically improve therapies exploiting HR. Current methods of assessing HRD are insufficient and consistently lead to an underestimation of the number of patients with HRD tumors (Table 1). Currently, the main application of HRD assessment in the clinic is to select patients benefitting from treatment with PARPis. The interaction between PARP inhibition and HRD can be described as synthetically lethal, where an individual loss of gene or protein of either is not lethal, but a combined loss of function results in cell death [81]. PARPis not only inhibit the catalytic function of PARP enzymes, preventing repair of single-stranded breaks (SSBs), but also trap PARP to sites of DNA damage, preventing DNA repair, replication, and transcription [82]. Ultimately, DSBs accumulate, which cannot be repaired in HRD cells. To date, three PARPis (olaparib, rucaparib, and niraparib) have been approved by the FDA for use in ovarian cancer patients with germline *BRCA2* mutations. Germline or somatic *BRCA2* testing is insufficient, however, and leads to an underestimation and even incorrect classification of patients who could benefit from treatment with PARPis. In addition to assessing germline or somatic *BRCA2* status, sensitivity to platinum-based agents is also used as a functional readout of HRD to select patients expected to be sensitive to PARPis [83]. However, mechanisms of inherent or acquired resistance to platinum-based agents extending beyond *BRCA1/2* status [84] also implicate a potential underestimation of patients that would benefit from PARPis. Complementary assessment of HRD scar would address this drawback and allow a more accurate identification of patients responsive to PARPis.

Supportive of this, patients with *BRCA1/2* wild-type platinum-sensitive ovarian carcinomas with high degrees of LOH responded better to rucaparib than patients with low degrees of LOH in the ARIEL2 phase 2 clinical trial (NCT01891344) [33], demonstrating the potential of PARP inhibition to be extended beyond *BRCA1/2* mutated tumors. Additionally, in the RUBY phase 2 trial (NCT02505048), evaluating the efficacy of rucaparib in metastatic breast cancer patients without germline *BRCA1/2* mutations, the HRDetect score tended to be higher in responders vs. nonresponders (0.465 vs. 0.040), albeit not significant [85], perhaps due to the low number of patients (*n* = 40) analyzed. In the PAOLA-1 phase 3 trial (NCT02477644), patients suffering from platinum-responsive ovarian cancer classified as HRD-positive based on the HRD score (≥42; Myriad myChoice^®^ CDx) demonstrated better response to olaparib even when no *BRCA1/2* mutations were present [86]. In this study, an additional 18% of *BRCA1/2* wild-type patients were identified as HRD and significantly benefited from PARP inhibition. Similarly, in the QUADRA trial (NCT02354586), also employing the Myriad myChoice^®^ CDx assay, 29% of patients were classified as HRD-positive without *BRCA1/2* mutations and responded significantly better to niraparib, especially when both HRD-positive and platinum-sensitive [87], implicative that combining genomic scar analyses and functional HRD assays might provide additional discriminative value of these biomarkers. Findings of these clinical trials are in line with a study by Davies and colleagues that demonstrated that up to 22% of breast cancer patients could potentially benefit from PARP inhibition as predicted by HRDetect, larger than the hitherto appreciated fraction (1–5%) [62] and highlighting the need for more accurate and integrated approaches for HRD classification.

### 7.2. Platinum-Based and Other DSB-Inducing Agents

In addition to PARPis, tumors deficient in HR are also more sensitive to other therapies inducing DSBs such as platinum-based agents [88]. These agents, including cisplatin and carboplatin, preferentially bind to the N7 atom of the purine bases of DNA, after which three types of lesions can form: monoadducts, intrastrand crosslinks, and interstrand crosslinks (ICLs), subsequently leading to SSBs or DSBs [89]. Given the complexity of DNA lesions induced by platinum-based agents, several DNA repair mechanisms are implicated in repair of the induced damage, including HR for DSBs [90]. Indeed, in several preclinical and clinical studies, cells and tumors with germline *BRCA1/2* mutations exhibit enhanced sensitivity to platinum-based agents [91,92,93,94]. However, it has previously been demonstrated that TNBC patients without *BRCA1/2* mutations but with either low *BRCA1* mRNA expression or promoter methylation also showed improved responses to cisplatin [95]. In line with this, the burden of TAI and LOH could successfully predict (AUC = 0.74) the response of breast cancer patients to cisplatin in two phase 2 clinical trials (NCT00148694 and NCT00580333), an association that remained significant when only wild-type *BRCA1/2* cases were included [30,34]. Additionally, in the PrECOG 0105 phase 2 trial (NCT00813956), clinical responses of wild-type *BRCA1/2* breast cancer to cisplatin patients were higher in patients classified as HRD defined as a LOH score ≥10 [4]. In this trial, 57% of wild-type *BRCA1/2* patients were classified as HRD and experienced significant additional benefit from carboplatin treatment [96]. In a different phase 2 trial (NCT01372579) in TNBC patients, a positive HRD score (≥42; Myriad myChoice^®^ CDx) successfully predicted response to carboplatin with 75% of responsive patients not harboring germline *BRCA1/2* mutations [97]. Conversely, in the TBCRC phase 2 trial (NCT01982448) in TNBC patients, no significant association was observed between the HRD score determined by the Myriad myChoice^®^ CDx assay and clinical response to cisplatin [98]. A different cutoff value to classify tumors as HRD-positive (≥33 vs. ≥42) used in this study, however, could be a potential explanation for this discrepancy.

Assessment of HRD could also be applied to other DSB-inducing therapies. Indeed, several preclinical studies observed enhanced sensitivity of HRD cells and tumor xenografts to ICL-inducing agents such as chlorambucil [99] and mitomycin C [100]. Clinical trials, comparing the efficacy of nonplatinum interstrand crosslinking agents in tumors proficient and deficient in HR, however, are limited. In a retrospective analysis of 83 TNBC patients in a TCGA cohort, patients classified as HRD by HRDetect (≥42) responded significantly better (overall survival; *p* = 0.0063) to the interstrand crosslinker cyclophosphamide in combination with anthracycline and taxane chemotherapy (ACT) [101]. Moreover, HRD patients were less likely to be resistant to ACT combinational treatment when compared to HR-proficient patients (37.1 vs. 62.9%; *p* = 0.074) [101]. Similar results were observed in the SWOG S9313 trial (Int0137), as HRD-positive status (HRD score ≥42) was associated with better disease free survival (DFS; hazard ratio 0.72; *p* = 0.049) after combination treatment of cyclophosphamide and anthracycline (AC) [102].

Taken together, results of these clinical trials underscore that application of a reliable HRD biomarker could improve prediction of treatment response to DSB-inducing agents, and might lead to optimization of treatment regimen selection, not only of existing and approved drugs, but also of novel, more targeted therapies. Hypoxia-activated prodrugs (HAPs), for example, are a promising therapeutic approach selectively targeting hypoxic tumor cells associated with malignant progression and resistance to conventional therapies [103]. Despite highly promising preclinical and clinical results [104,105,106,107], HAPs have failed to be implemented in routine clinical settings with a lack of patient stratification partly accountable for their failure [105]. Identification of key factors of the tumoral response to HAPs, and predictive biomarkers thereof, is thus essential for their successful clinical application. In this light, Hunter and colleagues demonstrated that for several hypoxia-activated alkylating agents, including evofosfamide and PR-104, cytotoxicity and antitumor effects were markedly enhanced (*p* < 0.0001) when cells and tumors were HRD [108]. CP-506, a next-generation hypoxia-activated alkylating agent, demonstrated highly hypoxia-selective cytotoxicity and induction of DNA interstrand crosslinks [107,109]. Furthermore, the antitumor effects of CP-506 were significantly enhanced in HRD tumors (unpublished data). Based on these data, CP-506 will be evaluated in an upcoming pan-cancer phase 1/2 clinical trial TUMAGNOSTIC (NCT04954599) in solid tumors with HRD or in tumor types with a high incidence of HRD. In this trial, CHORD will be used for the detection of HRD as it is better suited for HRD detection in tumor-agnostic clinical trials.

### 7.3. Immunotherapies

The integrity of DNA repair pathways can modulate the immune system and antitumor immunity—and therefore also response to immunotherapies—in a variety of ways. Accumulation of mutations in the DNA due to elevated genomic instability, for example, can result in novel proteins normally not encoded, creating tumor-specific antigens called neoantigens. Several studies have demonstrated a correlation between HRD and tumor mutational burden (TMB) and neoantigen load, as reviewed recently [110]. A higher neoantigen load has been associated with both longer overall survival and increased response to immunotherapies in lung cancer patients [111]. Supportive of this, recent studies found a correlation between the HRD score and increased neoantigen load and TMB in a pan-cancer cohort [112], and demonstrated that HRD ovarian and prostate tumors exhibited increased lymphocyte infiltration [113,114,115].

Despite these immunogenic events, HRD tumors are able to evade immune clearance via modulation of the tumor microenvironment (TME) by release of immune-suppressive cytokines [116], and upregulation of immune checkpoint molecules programmed cell death 1 receptor (PD-1) and its ligand PD-L1 [117,118,119,120,121], and cytotoxic T-lymphocyte-associated protein 4 (CTLA-4) [117], effectively promoting an immunosuppressive TME. In support of these findings, Jenzer and colleagues found that, although tumors with a loss of *BRCA2* showed increased levels of tumor-infiltrating lymphocytes (TILs), the ratio of cytotoxic CD8^+^ T lymphocytes to immunosuppressive T regulatory CD25^+^ FOXP3^+^ lymphocytes was lower compared to *BRCA2* wild-type tumors [113]. This highly immunosuppressive TME of HRD tumors, however, also renders them more sensitive to immune checkpoint inhibitors (ICIs). In a retrospective analysis of three clinical trials assessing monotherapeutic efficacy of anti-PD-1/PD-L1 in advanced urothelial cancers, alterations in HR genes were more commonly observed in responders compared to nonresponders [122]. Additionally, *BRCA2* mutations were significantly enriched in melanomas responsive to anti-PD-1 therapy [123].

Of note, it is important to distinguish between *BRCA1*- and *BRCA2*-mutated tumors regarding their immunophenotype and clinical effectivity of ICIs. In high-grade ovarian cancer, tumors with alterations in *BRCA1*, but not *BRCA2*, for example, demonstrated a more immunoreactive phenotype characterized by higher levels of TILs [124]. In breast cancer, on the other hand, a more immunosuppressive phenotype of *BRCA1*-mutated tumors was observed compared to *BRCA2*-mutated tumors [125]. This is further reflected by several preclinical studies showing that *BRCA2*-mutated, but not *BRCA1*-mutated breast cancers, are responsive to treatment with ICIs [117,125]. Prospective studies are thus needed to assess whether HRD and HRD-associated scars can be used as biomarkers of response to treatment with ICIs.

## 8. Conclusions and Future Directions

HRD is a well-recognized characteristic of tumors and is associated with an enhanced sensitivity to several anticancer therapies. Accurate detection of clinically relevant HRD would allow proper patient stratification and improve the outcome of conventional and targeted therapies such as PARPis, platinum-based and other DSB-inducing agents, immunotherapies, and potential combinations thereof, which warrants further investigations. However, current clinical assessment of HRD—mainly relying on determining germline *BRCA1/2* mutational status—is insufficient and consistently leads to an underestimation of the number of patients with HRD tumors as mechanisms of HRD occurrence extend beyond functional BRCA1/2 loss. HRD, regardless of *BRCA1/2* status, is associated with specific forms of genomic and mutational signatures termed HRD scar. A variety of these scars have been identified and associated with HRD, including large-scale genomic aberrations such as TAI, LOH, and LST, and mutational signatures such as SBS3 and ID6. Nonetheless, composite models—such as the HRD score, HRDetect, and CHORD—integrating multiple of these scars, better encompass the full complexity of HRD-associated phenotypic changes and are in all likelihood better candidates for clinical application for detection of HRD. Last but not least, integrating genomic (scar) and more functional repair analyses—either clinical response data or functional HRD assays—likely further improve the prognostic and predictive value of these biomarkers.

## Figures and Tables

**Figure 1 cancers-14-04157-f001:**
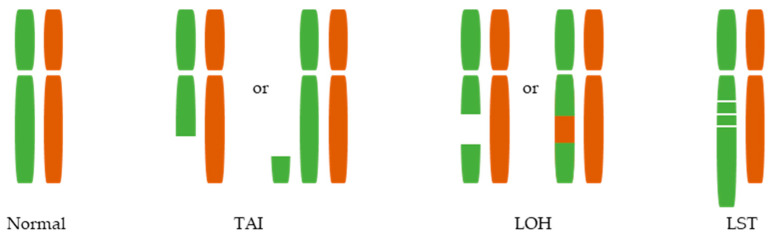
Large-scale genomic aberrations associated with homologous recombination deficiency. At least three large-scale genomic aberrations have been associated with HRD: telomeric allelic imbalance (TAI), defined as regions of allelic imbalance at telomeric region; loss of heterozygosity (LOH), defined as the loss of one or two alleles at a locus; and large-scale transitions (LST), defined as chromosomal breaks between regions of at least 10 Mb.

**Table 1 cancers-14-04157-t001:** Overview of completed clinical trials to date determining patient HRD status by assessing the presence of HRD scars.

Name	Phase	Cancer Type	Patients	HRD Assay	*BRCA1/2* Mutation	Total HRD	Aims and Results	Study Identifier	Associated Publications
PrECOG 0105	2	Breast	80	LOH score	24.2%	81.5%	Assessment of the safety and efficacy of iniparib in combination with gemcitabine and carboplatin. Mean LOH score was higher in responders vs. nonresponders, an association that remained significant when only *BRCA1/2* wild-type tumors were considered.	NCT00813956	Telli et al., *J. Clin.* *Oncol.* **2015**.
NU 10B07	2	Breast	30	HRD score	10.0%	46.2%	Evaluation of safety of carboplatin and eribulin in breast cancer patients and use of HRD score as biomarker of response. Combination of carboplatin and eribulin was safe and HRD score could predict outcome regardless of *BRCA1/2* mutational status.	NCT01372579	Kaklamani et al., *Breast Cancer Res. Treat.* **2015**.
ENGOT-OV16/NOVA	3	Ovarian	594	Myriad myChoice^®^ CDx	36.7%	78.5%	Evaluation the efficacy of niraparib in platinum-sensitive ovarian cancer. Niraparib treatment resulted in significant longer progression-free survival in patients with *BRCA1/2* mutation and positive HRD classification.	NCT01847274	Mirza et al., *NEJM* **2016**.
ARIEL2	2	Ovarian, fallopia, peritoneal	206	LOH score	19.6%	59.2%	Evaluation of LOH as biomarker of response to rucaparib. Patients without *BRCA1/2* mutation but high LOH score responded better to rucaparib.	NCT01891344	Swischer et al., *Lancet Oncol.* **2017**.
Study 19	2	Ovarian	53	HRD score	60.3%	69.8%	Characterization of long-term and short-term responders to olaparib. *BRCA1/2* status and a high HRD score were associated with long-term response to olaparib.	NCT00753545	Lheureux et al., *Clin. Cancer Res.* **2017**.
SWOG S9313	2	Breast	425	HRD score	29.3%	67.3%	Evaluation of the combination of anthracycline and cyclophosphamide in breast cancer patients and use of HRD as biomarker of response. HRD-positivity was associated with better response to anthracycline and cyclophosphamide combination.	Int0137	Sharma et al., *Ann. Oncol.* **2018**.
M10-976	1	Ovarian, fallopia, peritoneal	60	HRD score	43.3%	56.7%	Assess the safety of a novel PARPi, ABT-767. Patients with a mutation in *BRCA1/2* or with a HRD score ≥42 responded better to ABT-767.	NCT01339650	Van der Biessen et al., *Invest. New Drugs* **2018**
QUADRA	2	Ovarian, fallopia, peritoneal	463	Myriad myChoice^®^ CDx	18.7%	47.9%	Assessment of safety of niraparib in patients with ovarian, fallopian, or peritoneal cancer.	NCT02354586	Moore et al., *Lancet Oncol.* **2019**.
PAOLA-1	3	Ovarian, fallopia, peritoneal	806	Myriad myChoice^®^ CDx	30.0%	48.0%	Assessment of efficacy of the combination of olaparib and bevacizumab in ovarian cancer. Significant benefit of addition of olaparib was observed in patients with a high HRD score, regardless of *BRCA1/2* mutations.	NCT02477644	Ray-Coquard et al., *NEJM* **2019**.
LIGHT	2	Ovarian, fallopia, peritoneal	272	Myriad myChoice^®^ CDx	37.0%	62.2%	Evaluation of safety of olaparib in patients with ovarian, fallopian, or peritoneal cancer and use of HRD score as biomarker of response. Patients with *BRCA1/2* mutations and positive HRD score responded better to olaparib.	NCT02983799	Cadoo et al., *2020 ASCO Annu. Meet.* **2020**.
GeparOla	2	Breast	107	HRD score	56.2%	99.7%	Investigation of the combination of paclitaxel and olaparib (PO) in HER2-negative HRD breast cancer patients. PO was safe and resulted in higher pathologic complete response.	NCT02789332	Fasching et al., *Ann. Oncol.* **2021**.
RUBY	2	Breast	40	LOH score HRDetect	12.5%	100.0%	Assessment of efficacy of rucaparib in breast HER2-positive HRD breast cancer and use of LOH and HRDetect as biomarkers of response. A positive HRD score was an inclusion criterion. A subset of patients with high LOH and HRDetect scores without *BRCA1/2* mutations benefited from rucaparib treatment.	NCT02505048	Patsouris et al., *EJC* **2021**.
TBCRC	2	Breast	138	Myriad myChoice^®^ CDx	6.7%	71.2%	Assessment of a correlation between HRD score and response to cisplatin or paclitaxel. Tumors with a HRD score ≥33 were classified as HRD instead of ≥42. HRD was not predictive of response to either cisplatin or paclitaxel.	NCT01982448	Mayer et al., *Ann. Oncol.* **2021**.
Lung-MAP S1900A	2	Lung	59	LOH score	38.9%	100.0%	Assessment of efficacy of rucaparib in lung cancer patients with HRD. A positive LOH score was an inclusion criterion. The degree of LOH did not predict response to rucaparib.	NCT03377556	Riess et al., *2021 ASCO Annu. Meet.* **2021**.
JBCRG-22	2	Breast	99	HRD score	N/A	46.5%	Investigation of clinical usefulness of combination of eribulin and carboplatin or paclitaxel. HRD patients (≥42) responded significantly better to combination therapy.	UMIN000023162	Masuda et al., *Breast Cancer Res. Treat.* **2021**.
Meet-URO 12	2	Urothelial	58	FoundationOne^®^ CDx	12.8%	44.7%	Evaluation of niraparib in combination with best supportive care (BSC). Addition of niraparib did not improve progression-free survival	NCT03945084	Vignani et al., *2022 ASCO Genitourin.* *Cancers Symp.* **2022**.

HRD assay: type of assay used to detect presence of HRD scar; *BRCA1/2* mutation: percentage of patients in the trial with a *BRCA1/2* mutation; Total HRD: total percentage of patients with HRD, either a *BRCA1/2* mutation and/or positive HRD score; N/A: not applicable, i.e., not reported.

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
