# Peer review of "Homologous Recombination Deficiency Scar: Mutations and Beyond—Implications for Precision Oncology"

_cancers, 2022, doi:10.3390/cancers14174157_

Round 1

Reviewer 1 Report

Comments: van der Weil et al present a review titled Homologous recombination deficiency: mutations and beyond – implications for precision oncology. The authors review that tumor intrinsic alterations in HR and the associated HR deficiency phenotype is well recognized in cancer. Further, recognition of HRD has clinical implications, as tumors showing HRD have better responses to certain DNA damaging agents. The authors also describe how current testing strategies for HRD are insufficient, failing to fully capture the HRD patient population. The lack of robust testing is a significant challenge in the DNA damage repair deficiency field, making this review timely. To address this, the authors review novel and emerging testing strategies to identify HRD and describe implications for clinical oncology moving forward. The authors focus on genomic scar and mutational signatures. The main limitation is that this this topic has been frequently reviewed. However, the authors are very comprehensive, and I especially enjoyed the presentation of clinical trials which have incorporated scar/mutational signature testing.

Comments:

1.     The title (mutations and beyond) suggests a comprehensive review of testing strategies for HRD, yet the focus is only really on genomic scar and mutational signatures. As such, the authors should (i) expand the discussion on functional testing for HRD and discuss RAD51 foci analysis and any other emerging functional tests. (ii) It may also be worthwhile to comment on mutations in non-BRCA HR genes which associate with sensitivity to DNA damaging therapy. Lastly, (iii) clinical selection using sensitivity to platinum chemotherapy is also a commonly used selection criteria to identify patients eligible for PARPi, often in combination with germline/somatic testing for BRCA mutations. The authors should discuss whether clinical criteria need to be paired with scar/mutational signatures and identify when that was done in the clinical trials discussed in the section on PARPi.   

2.     The authors should discuss the limitations of genetic signature testing in regard to clinical use. Fresh tissue is often required as formalin fixation can induce artifact (Davies et al. Nature Medicine. 2017).

3.     The comment on SigMA is interesting. Are there other emerging computational approaches either to genomic analysis or pathology analysis that could be incorporated to identify HRD?

4.     Throughout the manuscript the authors comment that a common characteristic of tumors in HRD. Although HRD is well recognized, it is not necessarily common to many cancers, as HRD tumors still represent a minority of cancers. 

5.     For Figure 1, can the authors make it clearer that for TAI and LOH where two sets of chromosomes are depicted that the chromosome pairs are separate? Can the authors redraw LST to better depict chromosomal breaks? Also, consider including the definitions of genomic scar in the figure legend.

6.     ID6 should include a reference to Alexandrov. Nature. 2018.

7.     Hypoxia activated prodrugs section feels a bit out of place, as these are really just DNA damaging chemotherapies with a twist. As such, it would not be surprising that they would perform better in HRD tumors. Perhaps this should be included in section 6.2 or subsections for 6.2 generated.  

8.     The authors comment that germline testing for BRCA mutations is most common. However, somatic testing is also frequently performed and recommended by cancer societies for a variety of tumors.

9.     The explanation of PARPi mechanism of action is now thought to be ‘trapping’ of PARPi generating PARPi/DNA complexes, rather than simply inhibition of ssDNA breaks (Pommier. Science Trans. Medicine. 2016).

Author Response

Dear reviewer,

We would like to thank you for your time to read and review our manuscript – Homologous Recombination Deficiency Scar: Mutations and Beyond – Implications for Precision Oncology.

Attached, you can find our responses to your comments in red. Changes in the manuscript and figures according to your comments are also indicated in the respective document in red.

Reviewer 2 Report

In this report, the authors attempt to provide a review of HR deficiency signatures and how they are employed in precision medicine for cancer treatments. The authors provide some literature evidence for their review, but the integration of the literature is poor. First, the authors focus primarily on BRCA1/2 mutations even though their goal in the title and abstract is to review HR deficiency which goes beyond BRCA1/2 mutations. Second, the authors fail to adequately describe the role of HR (they assume that all HR is error free and that it only works in the absence of NHEJ). Further, the role of RAD52 which currently actively investigated and even used as a genetic marker is omitted. Other suggestions I point below in my major revision section.

However, I believe that there is some merit in this paper, but I suggest the following major changes in addition to my comments below:

1.       Include a section on HR and NHEJ from a genetic point of view: e.g. Describe the players in HR and NHEJ with some diagrams.

2.       Include a section on historical findings of HR deficiencies and their connection with chromosomal re-arrangements. Also talk about how these historical findings (e.g. Philadelphia chromosome) have been used as clinical markers.

3.       Include a section on other HR related genes and their uses in clinical diagnosis (e.g. checkpoints, chromatin remodelers, etc).

4.       Discuss major findings in genome wide chromosomal re-arrangement analysis as potential signatures (I point out chromothripsis, etc below). Here is a review on re-arrangements (PMID: 35091282).

As it stands now, the paper is too narrow and does not really address how HR deficiency is used clinically for diagnosis.

Major revisions

1.       Paragraph at lines 60-66. The statement in this paragraph is not completely true. In fact, there is evidence that NHEJ works throughout the cell cycle while HR functions primarily in S phase and G2/M. Thus, to say that NHEJ is used to repair DNA damage in the absence of HR is incorrect. NHEJ is uses even in the presence of HR. Please review literature on the cooperation/competition between HR and NHEJ as well as their role in the cell cycle. Further, not all HR is error free. Take for example SSA which can produce intra-chromosomal deletions or BIR which can produce non-reciprocal translocations.

2.       Paragraph starting at line 67. The statement here that HRD was associated with TAI was the first thing discovered is not accurate. Chromosomal translocations ware observed way before that and correlated with deficiencies in HRD (e.g. Bloom syndrome, PMID: 510071; XP, FA and AT, PMID: 258177; Philadelphia chromosome, PMID: 2581635; MSI in colorectal cancers PMID: 8700523…and the list can go on…). The Philadelphia chromosome definitely qualifies as a large-scale genomic aberration. Why are the authors not mentioning these early seminal studies? It appears that the authors focus primarily on BRCA1/2 related chromosomal re-arrangements. But if the goal is to provide a review of studies that show that HR deficiency cause re-arrangements then these early seminal findings must be included.

3.       The section describing mutation signatures in HRD. The authors fail to include major signatures such as chromothripsis (PMID: 21215367, also see PMID: 21215363 and PMID: 35803217 for comment on chromothripsis and historical view of re-arrangements), chromoplexy (PMID: 23680143) and break fusion bridge cycles (PMID: 32299917).

4.       The roles of RAD52 (for which single molecule inhibitors are being developed), chromatin remodeling proteins, checkpoint proteins as well as other genes such as metabolic genes (e.g. IDH1) are not discussed. For example, the author should be aware that screening for RAD52 in certain BRCA1/2 mutated cancers is a current clinical focus as repair relies on RAD52 and can produce translocations. Chromatin remodeling factors such as KAT5 and PRMT5 are currently also screened for. Finally, mutations in metabolic genes such as IDH1 is a major screening method especially in CNS cancers (PMID: 34503108). Even more importantly, mutations in IDH1 have been linked to HDR deficiencies (PMID: 32494005). Why are the authors ignoring these studies?

Minor revisions

1.       Line 13: use “homologous recombination deficiency” so that it would fit with the HRD acronym.

2.       In the field of DNA damage repair genetics LST is somewhat vague. Please further define LST in terms of translocations or copy number variations.

Author Response

(The authors gave the same response as above.)

Round 2

Reviewer 2 Report

The authors have made significant changes or provided compelling arguments. This reviewer is satisfied.